# Investigating the limits of free-form debate as a scalable oversight strategy

**Gareth Tan**                                                    *gareth-ais@proton.me*
*MARS, Cambridge AI Safety Hub*

**Leonid Tsyplenkov**                                          *LeonidTsyplenkov@gmail.com*
*Wentworth, University of York*

**Edy Nastase**                                                *tollymon2017@gmail.com*
*University of Galway*

**Gabriel Recchia**                                           *gabe@moduloresearch.com*
*Modulo Research*

**Reviewed on OpenReview:** *https://openreview.net/forum?id=vRCGzuAhOM*

## Abstract

Debate is a scalable oversight method involving two copies of a strong model trained to defend alternative responses to a question, with a judge with less task-relevant information, time, or domain-specific capability evaluating which answer is better supported. We replicate and extend a result from prior work demonstrating that training Llama3-8B-Instruct-262k as a debater led to increased performance of a GPT-4-class judge model on QuALITY, a question-answering task that grants the debaters a capability advantage via information asymmetry. When replicating the original setup as closely as possible, we confirm that training debater models in free-form, multi-round debate increased judge accuracy. However, this finding did not generalize across alternative tasks or models, and did not replicate consistently under our closest approximation to the original setting. These results suggest that the effectiveness of free-text debate as a scalable oversight method is sensitive to task structure, model pairing, and training conditions, and highlight the need for greater understanding of when and why debate improves judge accuracy. We identify several factors that may influence debate's success and outline directions for future work aimed at characterizing the conditions under which debate strengthens oversight reliably.

## 1 Introduction

Advances in AI capability are increasing the cost and difficulty of verifying the accuracy of model outputs. This problem is particularly acute in domains requiring specialized knowledge or complex reasoning, where even experts may struggle to verify whether an AI's response is correct. Without scalable oversight mechanisms, we risk deploying increasingly powerful systems whose outputs we cannot reliably assess. One potentially promising mechanism is *debate* (Irving et al., 2018; Buhl et al., 2025). Rather than requiring a human or model evaluator to verify complex outputs directly, debate leverages adversarial dynamics: two systems argue for opposing answers while a judge with less task-relevant information, time, or domain expertise determines which answer is better supported. The key insight is that it may be harder for a strong model to hide a flaw in its reasoning than it is for another copy of it to surface that flaw to a judge, enabling the judge to leverage the power of a very capable model to evaluate its own outputs. If this approach works, it could enable humans to oversee AI systems in domains where direct evaluation is infeasible.

Recent work by Arnesen et al. (2024) provided the first empirical evidence that training models to argue their cases more persuasively through self-play can result in increased judge accuracy in a natural language debate setting. They used QuALITY (Pang et al., 2021), a task in which judges must answer questions about source texts that they do not have access to, but which are included in the debaters' context windows. This creates a controlled information asymmetry: the debaters have access to evidence that the judge cannot inspect directly. We use this setup as a tractable proxy for one aspect of scalable oversight, namely cases in which an evaluator must assess claims whose full evidential basis is unavailable or too costly to inspect. It should not be read as implying that the debaters are globally more capable than the judge, or that future oversight systems should prefer stronger debaters without also improving judges. Rather, the setup isolates whether adversarial presentation of selectively quoted evidence can help a comparatively information-limited judge identify the correct answer. QuALITY rewards debaters with stronger abilities to select and repeat quotes from a source text which appear to support the debater's favored answer. If the strong long-context abilities of modern language models continue to scale linearly with model size (Hamilton et al., 2025), this task may provide limited headroom for evaluating how scalable oversight techniques scale as models become more capable. Convincing arguments for QuALITY questions generally do not require the construction of multi-step arguments involving reasoning or problem-solving. Moreover, Arnesen et al. (2024)'s experiments used relatively weak models as debaters (a fine-tuned version of Llama3-8B), leaving open the question of whether debate training remains robust when deployed with more capable models that might discover sophisticated strategies to mislead judges.

We independently replicate a key finding of Arnesen et al. (2024) and address these limitations through two extensions. First, we develop a novel task based on CELS-Lojban (Recchia et al., 2025), which includes questions requiring reasoning over a complex ruleset. Second, we scale up to stronger models by fine-tuning *o4-mini*, allowing us to investigate whether judge accuracy continues to improve after debate training when a more capable model is used as the debater. Together, these extensions test whether the benefits of training models in free-form debate observed by Arnesen et al. (2024) generalize to alternative tasks and to settings with more powerful debaters.

Our results reveal both promise and challenges for debate as a scalable oversight method. We find that debate training improved judge accuracy on QuALITY under conditions closely matching those of Arnesen et al. (2024), but that this improvement did not generalize to our reasoning-based task or to settings using stronger debater models. Moreover, the original finding proved difficult to replicate consistently, with judge accuracy varying substantially across training runs. We consider several factors that may explain this variability and discuss implications for future work on debate as an alignment strategy.

## 2 Related Work

**Scalable Oversight.** Scalable oversight concerns methods for enabling evaluators who are limited relative to the system under evaluation - for example by information access, available time, domain expertise, or model capability - to assess outputs more reliably than they could by direct inspection alone. Early proposals in this area include debate (Irving et al., 2018), recursive reward modeling (Leike et al., 2018), and iterated amplification (Christiano et al., 2018). Subsequent work has examined both extensions and failure modes of these approaches, including obfuscated arguments in debate (Barnes et al., 2020; Brown-Cohen et al., 2025), collaborations between weaker overseers and stronger models (Bowman et al., 2022; Cotra, 2021; Bowman, 2022), prover-verifier games (Anil et al., 2021; Kirchner et al., 2024), and the use of model-written critiques to assist evaluation (Saunders et al., 2022). More recently, researchers have also begun to explore unsupervised oversight methods for settings in which high-quality human labels are unavailable or too expensive to obtain. For example, Wen et al. (2025) study an unsupervised post-training method that fine-tunes models on labels derived from the models' own outputs rather than from human supervision. This line of work is relevant because, like scalable oversight more broadly, it addresses how reliable judgments can be elicited when direct supervision is weak or unavailable.

**Debate.** Debate is a scalable-oversight proposal in which competing agents argue for different answers to a question, with the goal that a weaker judge can identify the correct answer by evaluating the exchange (Irving et al., 2018). Like Arnesen et al. (2024), Radhakrishnan (2023) explored RL training of debaters,

but found little evidence that debater Elo was correlated with judge accuracy, speculating that this may have been due to the limited capability levels of the models evaluated. There is considerably more empirical work on inference-time optimization of both human and AI debaters. Michael et al. (2023) compared human debaters to AI debaters and found that higher debater skill leads to better judge accuracy. Khan et al. (2024) found that judge accuracy in debate is higher than in closed consultancy for QuALITY. While this extends to extractive QA tasks, debate does not consistently provide the same boost over directly asking the judge to solve tasks without information asymmetry (Kenton et al., 2024). Although QuALITY has been widely used to instantiate information asymmetry in debate studies, Adhikari & Lapata (2025) use multimodal debaters with text-only judges and find that debate reliably outperforms closed consultancy. In contrast to prior debate work, we study debate training on a more complex information-asymmetric task requiring reasoning over a large Lojban ruleset, and we also evaluate whether judge accuracy improves when stronger debaters are used on reasoning-heavy question-answering tasks.

## 3 Methods

### 3.1 Replication of key results from Arnesen et al. (2024)

We sought to replicate Arnesen et al. (2024)'s key finding that training debaters via self-play improves judge performance compared to using untrained debaters. The following sections describe the data sources, debate format, and training procedures for debater and judge models.

#### 3.1.1 Data sources

The primary dataset for this study is based on the *QuALITY* benchmark (Pang et al., 2021), specifically the *QuALITY-HARD* subset containing long-context reading comprehension questions over short stories. We followed Arnesen et al. (2024) in filtering this subset to include only those questions paired with a single correct answer and one "best distractor" as determined by human annotators. Debate transcripts for model training were compiled from two sources. First, like Arnesen et al. (2024), we used 237 debates between human debaters and judges from Michael et al. (2023). For alignment with our debate processing pipeline, each debate was segmented into multiple shorter debates when possible: if a transcript contained four speeches per side, we generated four separate debates by progressively adding pairs of speeches to the debate history. This approach increased the number of training items while preserving turn-by-turn coherence.

Second, we employed LLM debate transcripts from Khan et al. (2024). Because the exact transcripts that Arnesen et al. (2024) obtained from this study were no longer available, we used the outputs from Khan et al. (2024)'s Experiment 8 as a substitute. We randomly selected 238 debates from this subset. These two sources were combined to form the complete dataset for supervised fine-tuning of the judge and debater models.

#### 3.1.2 Debate format and preprocessing

Like Arnesen et al. (2024), we used a two-turn simultaneous debate format in which the debaters independently produced opening arguments, followed by a single rebuttal (four arguments in total). Each argument was represented in the training data as a separate prompt/completion pair. The prompt contained the *QuALITY* story, the question, the debater's assigned answer, and all previous speeches available at that point in the debate. We then applied a preprocessing step in which verified quotations from the source story were wrapped in `<quote>...</quote>` tags, while quotations that debaters wrapped in `<quote>` tags but which did not actually appear in the source were wrapped with `<invalid_quote>...</invalid_quote>`. For judge prompts, the full story was removed so that the judge only saw the question, the answer options, and the debate transcript (including quote-validation markup). Debater order was randomly permuted to mitigate position bias. Examples exceeding the maximum context length were removed to avoid truncating either the source story or the debate history.

### 3.1.3 Debate model selection and fine-tuning

Using the processed debate dataset, we performed initial supervised fine-tuning (SFT) of the debater model with the same underlying model and training pipeline as in Arnesen et al. (2024). Specifically, we used the *gradientai/Llama-3-8B-Instruct-262k* model checkpoint as our starting point. Training was conducted via the open-source codebase provided by Arnesen et al. (2024), adapted for our multi-turn debate format. The full configuration is provided in Appendix B. The dataset combined our debate corpus with the Alpaca-cleaned instruction-following dataset (Taori et al., 2023) to preserve general instruction adherence, following the hybrid fine-tuning approach also applied in Arnesen et al. (2024). The `max_length` parameter was set to 32,986 tokens to leverage the extended context capacity of the model.

### 3.1.4 Judge model selection and fine-tuning

For the judge model, our dataset mirrored the setup of Arnesen et al. (2024). We incorporated balanced role shuffling to ensure that Debater A and Debater B each won in exactly 50% of cases.

We conducted initial experiments using a fine-tuned *gpt-4.1* judge and configurations similar to the one presented in Appendix C. For comparability to Arnesen et al. (2024), we report the result of an experiment using 2,502 samples, in line with the operational test script in Arnesen et al. (2024)'s repository. We then conducted follow-up experiments using *gpt-4-turbo*, *gpt-4.1*, and *Llama-3-8B-Instruct-262k*[1] and a minimum of 2,603 samples[2] each, using configurations analogous to the example shown in Appendix D. In these follow-up runs, we evaluated *gpt-4.1* judges with and without SFT, and *Llama-3-8B-Instruct-262k* judges with and without SFT. As of the time of our experiments, SFT was not possible on the public version of *gpt-4-turbo*, so we could not reproduce Arnesen et al. (2024)'s exact judge setup for that model; we therefore treated the fine-tuned version of *gpt-4.1* as the nearest substitute.

The SFT procedures differed by model family. For Llama, both debater and judge models were trained locally using the Arnesen et al. supervised fine-tuning pipeline. The debater SFT started from *gradientai/Llama-3-8B-Instruct-262k*. The Llama judge was also trained with SFT rather than DPO, using the same repository pipeline (`run_sft.py`) but with judge-specific supervision. Concretely, judge training used full debate-transcript-to-judgment examples, where the model received the judge instructions together with the full debate transcript and was trained to produce a final decision in the format "Debater A | 90%". Thus, the judge was not trained as an open-ended debater, but as a conditional decision-maker over complete debate transcripts. For *gpt-4.1*, we likewise trained judge models separately with SFT for use in the main and follow-up evaluations.

Not all judge settings admitted fully matched pre/post debater comparisons. In particular, for the SFT-trained Llama judge we report evaluation with the round-two DPO debater, but we do not have a corresponding SFT-only-debater / SFT-trained-Llama-judge baseline in the follow-up comparison set. The output format was as in Arnesen et al. (2024), with predictions in the form "Debater X | Y%" where X is the predicted winning debater and Y is the confidence score (50-100%).

### 3.1.5 DPO training

DPO was applied to the debater, not to the judge. The judge supplied the preference signal used to score alternative debater speeches, but its parameters were not updated during DPO training.

We followed the high-level two-iteration DPO regime of Arnesen et al. (2024). Arnesen et al. report running two DPO iterations, adding 7,512 preference pairs in each iteration from both sides of 1,252 unique QuALITY training questions, with three preference pairs per round and two rounds per question. They trained low-rank adapters on the attention and MLP projection matrices with rank 128, using a mini-batch size of 32, a KL penalty of 0.5, reward scaling $\gamma = 7$ for debate, and SFT loss weight $\alpha = 0.005$. Our replication used the same two-round structure, the same number of preference pairs per iteration, the same LoRA rank, and the same reward-scaling and SFT-loss settings.

---

[1]The full pinned model names of the OpenAI models used are *gpt-4-turbo-2024-04-09* and *gpt-4.1-2025-04-14*, respectively.
[2]All comparison groups in QuALITY experiments using configurations similar to that in Appendix D used 2,604 samples, with the exception of three using 2,603 samples, and one case in which 2,796 samples were taken in error.

Using the fine-tuned debater and judge from the second step, we generated the first-round preference dataset for debater DPO training using the branching rollouts method described in the paper. For the first round of training, we used the hyperparameters described in the paper: a learning rate of $10^{-5}$, a KL penalty of 0.5, the scaled reward gamma of 7, and the SFT loss alpha of 0.005.

For the second round, we generated further preference pairs from the round-one DPO model and combined them with the first-round preference data to form the aggregate dataset used for the second DPO iteration. Following Arnesen et al. (2024), we initially attempted a learning rate of $5^{-5}$ for this second round. However, we observed negligible gradient updates to the LoRA adapters under this configuration, so we reverted to $10^{-5}$. The remaining core hyperparameters were kept the same: KL penalty $\beta = 0.5$, reward scaling $\gamma = 7$, SFT loss weight $\alpha = 0.005$, and LoRA rank 128 on the attention and MLP projection matrices.

## 3.2 Extending to stronger debaters

To examine whether increasing the base strength of the debater leads to higher overall accuracy, we additionally conducted reinforcement fine-tuning (RFT; OpenAI, 2025) of OpenAI's *o4-mini* model in the debater role. Whereas the supervised fine-tuning described in the previous section followed Arnesen et al. (2024)'s open-source pipeline and targeted Llama-based checkpoints, the present experiment adapts OpenAI's reinforcement fine-tuning API to the debate setting. This provided a controlled way to test whether a stronger proprietary model, once specialized as a debater, generalizes more effectively than open-source baselines. However, it was subject to a technical constraint from the OpenAI platform: instead of the self-play pipeline used to train *Llama-3-8B-Instruct-262k*, *o4-mini* could only be trained with RFT.

### 3.2.1 Training file construction

RFT on the OpenAI platform imposes a stricter data format than supervised fine-tuning. Each training example must consist of a `messages` array ending in a `user` role, with system messages disallowed. To satisfy these constraints, we designed a dedicated preprocessing pipeline that reformatted all debate transcripts previously used for fine-tuning the Llama model into reinforcement-compatible JSONL files. Each prompt comprised a single user message that inlined both the original system instructions and the debate context. This ensured that every prompt contained (1) the QuALITY story and debate question, (2) the two answer positions to be defended, and (3) all preceding speeches up to the current turn. For later turns, the accumulated transcript was progressively expanded to capture the full debate history.

In addition to the policy prompts, each item contained auxiliary fields for use by the grader: the debate topic, the two positions, the identity of the model's assigned role (Debater A or B), the most recent opponent speech, and a trimmed debate history excluding that final opponent turn. This separation allowed the debater model to be trained solely on the policy prompt while enabling the grader to reconstruct a complete view of both sides of the current round.

### 3.2.2 Grader design

Since fine-tuned models cannot currently serve as graders within the RFT framework, we tested *gpt-4-turbo*, *gpt-4.1*, and the fine-tuned version of *gpt-4.1* as the judging models, and embedded our customized judging rubric directly in the system prompt. The grader was instructed to evaluate only the debate topic, the declared positions, and the debaters' speeches, without access to the underlying story. At each training step, the grader appended the opponent's latest speech (from the dataset) and the policy model's new speech (from rollout) to the trimmed history. It then produced a scalar probability $p$ that Debater A's position was correct, which served as the reinforcement reward signal. In cases where the policy model played Debater B, the score was automatically transformed so that $p$ still denoted the probability of Debater A being correct, maintaining consistency across examples.

### 3.2.3 Objective and evaluation

By training *o4-mini* to act as a debater under this reinforcement regime, we aim to test whether a stronger pretrained model, once aligned with the debate format, yields improved generalization when evaluated against

a held-out set of QuALITY-HARD debates. Specifically, this setup allows us to measure whether enhanced debater persuasiveness translates into more accurate judge decisions, extending the central finding of Arnesen et al. (2024) beyond open-source Llama checkpoints. The resulting experiments thus directly probe the generalizability of debate-based supervision to stronger foundation models.

### 3.3 Lojban task

Lojban is a constructed language designed for cultural neutrality and lack of ambiguity. Since its initial development in 1987, it has cultivated a small but dedicated community of speakers who use it for purposes ranging from casual conversation to literary translation. The CELS-Lojban benchmark (Recchia et al., 2025) contains claims about Lojban syntax made by GPT-3.5 and GPT-4, with each claim evaluated for accuracy by two human experts. Incorrect claims include expert explanations of the underlying issues. This task is challenging for many language models; in Recchia et al. (2025), only OpenAI's o3 matched human expert performance. For many questions, effective arguments need to go beyond simply quoting particular parts of a text as evidence, instead involving reasoning about sentence composition algorithms and definitions (Table 1).

Table 1: Example item from the Lojban task.

**Example Question from Lojban Task**

| Section | Content |
|---|---|
| Problem statement | Lojban is a constructed human language created by the Logical Language Group in 1987. Which of the following is written in valid, grammatical standard Lojban? 
 (A) `jmifa clupa ca le tu` 
 (B) `le jmifa ca clupa tu` |
| Proposed solution | 1. The correct answer is (B) "le jmifa ca clupa tu". 

 2. This is because in Lojban, the predicate (the verb or adjective) always comes immediately after the sumti (the noun or pronoun). 

 3. In option (A), "jmifa clupa" is the predicate, but it comes before the sumti "le tu", which is incorrect. 

 4. In option (B), "clupa" is the predicate and it comes immediately after the sumti "le jmifa", which is correct. 

 5. Therefore, (B) is the grammatically correct option. 

 6. Additionally, in Lojban, the order of the sumti can be changed without changing the meaning of the sentence. 

 7. So, "le jmifa ca clupa tu" and "tu ca clupa le jmifa" both mean the same thing. 

 8. However, in option (A), the order of the sumti cannot be changed without changing the meaning of the sentence. 

 9. Therefore, (B) is the better answer. 

 10. Overall, it is important to remember the proper order of sumti and predicate in Lojban, and (B) follows this rule correctly. |
| Query | **"Is Sentence 3 of the proposed solution correct or flawed?"** |

*Note.* The correct answer is that Sentence 3 is flawed: the problem is that `le tu` is not a well-formed *sumti*, because `le` is a nominalizer and requires a predicate in its scope. A successful debater must therefore do more than quote a surface-level rule;

it must explain the relevant grammar and support that explanation with evidence from the reference context. The ground-truth labels from Recchia et al. (2025) determine whether the judge's answer is correct.

We adapted CELS-Lojban to create an alternative task for this study. Like QuALITY, our task incorporates information asymmetry: debaters receive partial access to reference documents, portions of which can be quoted to a judge as ground-truth evidence. We constructed these documents by combining materials from multiple sources. The first consisted of definition lists for numerous words of various Lojban grammatical classes (*gismu, lujvo,* and *cmavo*), and a compilation of abbreviated root word forms (*rafsi*) commonly used in compound creation. The second was a reference text of approximately 800k tokens of excerpts from two books commonly used as Lojban learning resources (Cowan, 2016; la klaku & Lojban Wiki contributors, 2013), broken into 318 sections.

It was not possible to include these documents in debaters' context windows in full due to cost and context-window limitations. We therefore explored different approaches for selecting relevant context to provide as quotable ground-truth data. After testing various methods on a development set of 124 questions to ensure debaters would receive sufficient information to assist judges without revealing answers outright, we ultimately selected a simple algorithm illustrated in Figure 1. First, definitions (including argument structure and usage notes) were retrieved from the definitions list for each word $w$ in the Lojban sentences in the problem statement (see Table 1 for an example). Additionally, for each such word, the two sections of our reference excerpt having the highest *tf-idf* with that $w$ were retrieved (using L2 normalization between vectors) and appended to the retrieved definitions. The decision to retrieve two sections was based on multiple considerations, including the context length of the smallest debate model and performance in experiments involving the development set. Other methods evaluated included providing more context in the form of basic Lojban lessons, using all of the filtered definitions list instead of only the most relevant ones, and others, but empirically retrieving definitions and the two sections with highest *tf-idf* for each Lojban word in the problem statement seemed most promising. Additional detail is provided in Appendix A. The test set consisted of 1,630 questions created by reformatting claims from CELS-Lojban (Recchia et al., 2025). Only claims for which both expert annotators agreed on correctness or incorrectness were included.

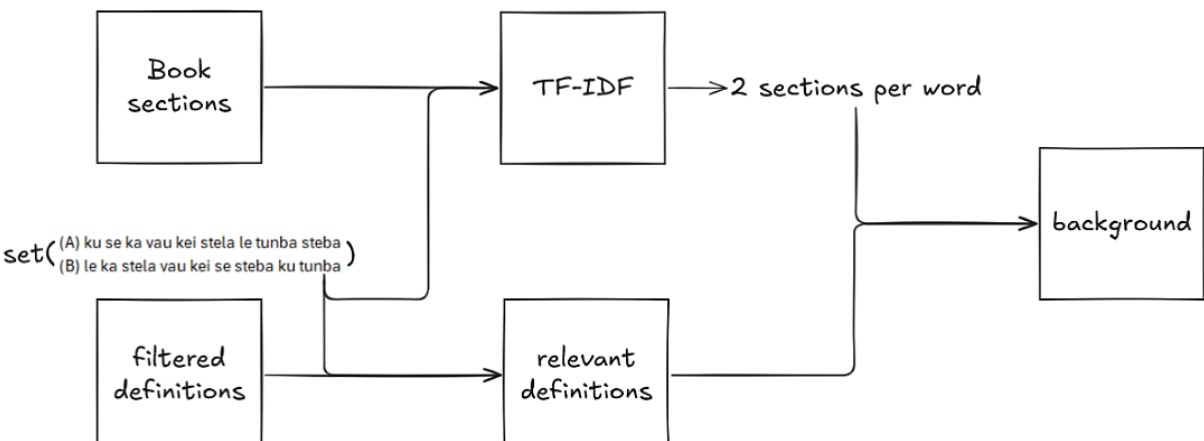

Figure 1: An overview of the algorithm used to extract the context about the Lojban language that was provided to debaters. This approach was used to extract context with a high likelihood of being relevant to each specific question under consideration.

## 4 Results

### 4.1 Replication of key results from Arnesen et al. (2024)

This section describes the results of the experiments from subsection 3.1.

Table 2: Effect estimates in the closest QuALITY replication.

| Outcome | Model | Estimate | Inference |
|---------|-------|----------|-----------|
| Judge correctness | Logistic mixed-effects model | OR = 1.39 | 95% CI [1.22, 1.57] |
| Judge correctness | GEE, clustered by item | OR = 1.20 | 95% CI [1.07, 1.35], $p = .002$ |
| Valid quotes per debate | OLS, clustered by item | +6.94 | 95% CI [6.58, 7.31], $p < .001$ |
| Invalid quotes per debate | OLS, clustered by item | +0.30 | 95% CI [0.21, 0.40], $p < .001$ |
| Total quotes per debate | OLS, clustered by item | +7.25 | 95% CI [6.87, 7.63], $p < .001$ |

*Note.* The judge-correctness models compare debates between SFT-only Llama debaters against debates between round-two DPO Llama debaters. GEE denotes generalized estimating equations, which estimate a population-averaged effect while accounting for repeated observations by item. Quote outcomes are mean differences after debate training. The evaluation used 2,502 examples under the configuration in Appendix C.

*Debater training resulted in debaters that were more capable of persuading GPT-4.1.* Across five experiments involving head-to-head debates over the 434 questions in our test set, the debater trained with SFT followed by DPO won about 15% more often than the debater trained with SFT alone. We estimated a logistic regression of the fully trained debater's wins on a centered indicator for whether the fully trained debater spoke first, with the intercept corresponding to the average advantage of the fully trained debater. Using two-way cluster-robust standard errors (clustered by question and experiment), the intercept was 0.314 (SE = 0.042, $z = 7.40$, $p < .001$), corresponding to an average win probability for the fully trained debater of 0.578 (95% CI [0.558, 0.598]). We additionally observed an apparent bias toward awarding the win to the first debater. This effect was positive but not statistically significant with clustering (log-odds = 0.284, OR = 1.33, 95% CI [0.87, 2.03], $p = 0.19$).

That said, we did not detect a significant association between training duration and debater performance, in contrast to Arnesen et al. (2024). In our case, regressing win probability on training time produced a slope statistically indistinguishable from zero. However, as we generated fewer than 1,000 debates for each checkpoint, it is possible that we had insufficient power to yield reliable point estimates for our 15 checkpoints. Additionally, Arnesen et al. (2024) used a different measure of win rate, namely average tournament-based Elo rather than head-to-head win probability.

*Training led to increased judge accuracy on QuALITY for GPT 4.1, but not consistently.* For the fine-tuned *gpt-4.1*-based judge, judge accuracy rose from 0.761 when judging debates between the SFT-only debaters to 0.790 when judging debates between the fully trained debaters. A logistic mixed-effects model including item as a random effect estimated an effect of training of 0.327 on the log-odds scale (SD = 0.065), corresponding to an odds ratio of 1.39. A population-averaged generalized estimating equation (GEE) clustered by item showed a smaller but still significant effect. For the *gpt-4.1*-based judge, using OLS with cluster-robust standard errors by item, the average number of valid quotes per debate rose from 11.75 before training to 18.69 after training, invalid quotes increased from 0.44 to 0.74, and total quotes rose from 12.18 to 19.43. Confidence intervals and other details about these analyses are reported in Table 2.

However, this positive result was not stable across nearby evaluation samples. The 2,502-sample comparison above was selected to match the operational evaluation size used in the original replication script. In addition to this reported subset, we inspected further outputs from the same larger evaluation stream by considering alternative subsets of 2,502 examples under the same model and debate settings. These alternative subsets did not show a comparable lift in judge accuracy after DPO training.

A near-identical follow-up run made this sensitivity more apparent. In this run, configuration values that had previously been inherited from defaults were written explicitly, and `shuffle_deterministically` was set to `True`, fixing the random seed used in dataset ordering. The debate format, models, prompts, and intended decoding procedure were otherwise kept the same. This run yielded accuracy of 0.774 with SFT-only debaters and 0.775 with round-two DPO debaters, a non-significant difference. We therefore interpret the change from 0.761 to 0.790 as evidence that debate training can improve judge accuracy in a close QuALITY replication, but not as evidence that this improvement is robust across nearby samples, orderings, or evaluation configurations.

Table 3: Judge accuracy before and after debate training across follow-up experiments.

| Task | Debater family | Judge | Judge training | Acc. before debate training | Acc. after debate training | Δ |
|------|----------------|-------|----------------|-----------------------------|----------------------------|---|
| QuALITY | Llama-3-262k | GPT-4-Turbo | Untrained | $0.681 \pm 0.009$ | $0.666 \pm 0.009$ | $-0.015$ |
| QuALITY | Llama-3-262k | GPT-4.1 | Untrained | $0.757 \pm 0.008$ | $0.747 \pm 0.009$ | $-0.010$ |
| QuALITY | Llama-3-262k | GPT-4.1 | SFT | $0.774 \pm 0.008$ | $0.775 \pm 0.008$ | $+0.001$ |
| QuALITY | Llama-3-262k | Llama-3-262k | Untrained | $0.512 \pm 0.010*$ | $0.500 \pm 0.010$ | $-0.012$ |
| QuALITY | Llama-3-262k | Llama-3-262k | SFT | $0.536 \pm 0.010*$ | $0.500 \pm 0.010$ | $-0.036$ |
| QuALITY | o4-mini | GPT-4-Turbo | Untrained | $0.725 \pm 0.009$ | $0.690 \pm 0.009$ | $-0.035$ |
| QuALITY | o4-mini | GPT-4.1 | Untrained | $0.815 \pm 0.008$ | $0.768 \pm 0.008$ | $-0.047$ |
| QuALITY | o4-mini | GPT-4.1 | SFT | $0.841 \pm 0.007$ | $0.797 \pm 0.008$ | $-0.044$ |
| Lojban | Llama-3-262k | GPT-4-Turbo | Untrained | $0.694 \pm 0.015$ | $0.666 \pm 0.012$ | $-0.028$ |
| Lojban | Llama-3-262k | GPT-4.1 | Untrained | $0.649 \pm 0.012$ | $0.640 \pm 0.012$ | $-0.009$ |
| Lojban | Llama-3-262k | GPT-4.1 | SFT | $0.575 \pm 0.012$ | $0.594 \pm 0.012$ | $+0.019$ |
| Lojban | o4-mini | GPT-4-Turbo | Untrained | $0.685 \pm 0.012$ | $0.679 \pm 0.012$ | $-0.006$ |
| Lojban | o4-mini | GPT-4.1 | Untrained | $0.771 \pm 0.010$ | $0.701 \pm 0.011$ | $-0.070$ |
| Lojban | o4-mini | GPT-4.1 | SFT | $0.742 \pm 0.011$ | $0.691 \pm 0.011$ | $-0.051$ |

*No other significant improvements.* Outside the closest operational replication subset, we did not observe statistically significant improvements in judge accuracy when comparing trained and untrained debaters. This includes nearby QuALITY follow-ups with the same Llama debater family and GPT-4-class judges, as well as experiments using `o4-mini` debaters and the Lojban task.

*Baselines.* To estimate overall capability on the question-answering task, adjusted for the information available to the debater and judge models respectively, the Inspect framework was used to directly prompt the "original" debater and judge models (e.g., the underlying models prior to SFT) with the questions from the test set. To mirror the circumstances of the debates, the original debater model (*Llama-3-8B-Instruct-262k*) was prompted with the story from QuALITY providing the necessary background to answer the question, as well as the question itself. The original judge model (*gpt-4.1*) was prompted with the questions alone, drawing 2,502 samples as in the main experiment. *Llama-3-8B-Instruct-262k* (with background) achieved accuracy of 0.623 (SE = .010), while *gpt-4.1* achieved a comparable 0.634 (SE = .010). Given that prior work found that inference-time debate is ineffective for eliciting improved performance when the same model is used as both judge and debater (Khan et al., 2024), it is interesting that we did observe statistically significant judge uplift after debater training for a debater and judge of comparable skill, albeit only for the configuration in Appendix C.

Arnesen et al. (2024) compare debate against consultancy baselines in which judges receive only a one-sided argument, which has a 50% chance of arguing for the correct answer. Here we concern ourselves only with the finding that judge accuracy increased after debaters underwent training.

*Note.* For Llama-based experiments, "before" denotes the SFT-only debater and "after" denotes the round-two DPO debater. For `o4-mini` experiments, "before" denotes the untrained debater and "after" denotes the RFT-trained debater. Asterisks mark Llama-judge baselines for which direct answering without the story outperformed self-debate with story access, so the direct-answering value is reported. Sample sizes were 2,604 for most QuALITY runs, 2,603 for the untrained `o4-mini` QuALITY runs, 2,796 for the Llama/GPT-4.1-SFT pre-training run, and 1,630 for Lojban, except for two Lojban Llama runs terminated early after it became clear that no improvement would be achieved. No row shows a statistically significant improvement after debate training.

## 4.2 Extending to stronger debaters

To test whether the judge-accuracy gains observed in our closest replication extend to more capable models, we fine-tuned `o4-mini` as a debater and evaluated the resulting debates on QuALITY. In this setting, we did not observe evidence that additional debate training improved judge performance. Instead, judge accuracy decreased for all three judges we tested. With an untrained `o4-mini` debater, judge accuracy was 0.725 for

GPT-4-Turbo, 0.815 for GPT-4.1, and 0.841 for the SFT-trained GPT-4.1 judge. After RFT, these values fell to 0.690, 0.768, and 0.797 respectively (Table 3).

This result is notable because the simple Inspect baselines show that `o4-mini` is very strong on the underlying QuALITY task when given access to the story, achieving 0.922 accuracy in the "untrained debater" setting. By comparison, the corresponding "untrained judge" baselines without access to the story were substantially lower: 0.588 for GPT-4-Turbo, 0.603 for untrained GPT-4.1, and 0.624 for the SFT-trained GPT-4.1 judge. These baseline results indicate that the setting still contains meaningful information asymmetry and that the debater has a substantial capability advantage on the task itself.

Taken together, these findings show that moving to a stronger debater did not make the judge-accuracy improvement more reliable in our experiments. In the `o4-mini` setting, additional debate training was associated with lower judge accuracy rather than higher judge accuracy across all three judges we evaluated.

### 4.3 Extending to the Lojban task

To assess whether debate training generalizes beyond extractive QA to tasks requiring complex reasoning, we evaluated trained debaters on the Lojban task. With the SFT-only Llama debater, judge accuracy on the Lojban task was 0.694 with the GPT-4-Turbo judge and 0.649 with the GPT-4.1 judge. These pre-training accuracies serve as baselines against which the effect of debate training on the out-of-distribution task can be assessed.

After DPO training, accuracy dropped with the GPT-4-Turbo judge (0.694 to 0.666) and the GPT-4.1 judge (0.649 to 0.640). The SFT-trained GPT-4.1 judge showed a small improvement (0.575 to 0.594). None of these differences were statistically significant, consistent with the QuALITY finding that debate training does not reliably improve judge accuracy. The lower accuracies observed on Lojban relative to QuALITY may reflect the task's complex multi-step reasoning demands, the fact that debaters were not trained on Lojban-related data, or both.

For `o4-mini` debaters, debate training was associated with decreased judge accuracy across all three judges, mirroring the pattern observed in QuALITY. The largest drop was observed with the untrained GPT-4.1 judge ($-0.070$). Comparing the pre- and post-training accuracies of the `o4-mini` baselines with those of Llama suggests that stronger base-model capability contributes to performance on this task. Taken together, debate training did not show improvements for either debater family on this task.

## 5 Discussion

Our findings present a mixed picture for free-form debate as a scalable oversight strategy. On the one hand, we reproduced a positive result in a close replication setting: training a Llama-based debater improved GPT-4.1 judge accuracy on QuALITY. On the other hand, this effect did not generalize across the rest of our experiments. A follow-up run using a nearly identical configuration did not show the same benefit, `o4-mini` debaters trained via OpenAI's fine-tuning API resulted in lower judge accuracy after training rather than higher, and results on Lojban likewise showed no significant improvements after debate training. The largest drop in accuracy was observed for `o4-mini` debaters with the untrained GPT-4.1 judge ($-0.070$).

Across the experiments, outcomes depended strongly on the model pairing and on the task. In QuALITY, we observed one clear positive replication in the Llama-based setting, but this pattern did not extend to `o4-mini`. For `o4-mini`, all three judges performed worse after debate training than before it. On Lojban, the post-training accuracies we observed were lower than the corresponding QuALITY accuracies for the same family of judge models. Taken together, these results do not support a simple claim that stronger debaters or additional debate training reliably improve judge accuracy.

Our results also suggest that raw task capability alone is not sufficient for robust judge improvement. The strongest simple baseline we measured was the untrained `o4-mini` debater on QuALITY, which achieved 0.922 accuracy when given the story. Nevertheless, training this model further for debate did not improve judge accuracy in the debate setting. In other words, strong performance on the underlying task did not automatically translate into better assistance for a judge lacking relevant information.

Judge choice also appears to matter. In the follow-up QuALITY experiments with Llama-based judges, judge accuracy was at or near chance, with accuracies ranging from 0.500 to 0.536 across the settings we evaluated. These runs also showed noticeable bias in favor of the first debater in some cases. More broadly, the difference between our positive replication and our near-null follow-up result indicates that debate outcomes can be sensitive to details of the experimental setup.

Overall, our experiments do not show that free-form debate is a robust, model-agnostic method for improving judge accuracy. Instead, they suggest that its effectiveness is conditional on the particular task, debater, judge, and training setup. The main positive result remains important, but our broader set of findings indicate that it should be interpreted as a setting-specific success rather than as evidence of a generally reliable approach.

Several directions follow from this. First, future work should use matched pre- and post-training comparisons for every task and model pairing. Second, trained debaters should be evaluated against a broader range of judges to test whether any gains transfer across evaluators. Third, it would be useful to study more systematically which properties of tasks and judge models are associated with stronger or weaker debate performance. Finally, given the sensitivity of the results across nearby configurations, more controlled comparisons will be necessary to identify which components of the debate pipeline are responsible for any observed gains.

## 5.1   Limitations and future work

This study has several limitations. Although we replicate one positive result under a configuration close to that of Arnesen et al. (2024), our follow-up experiments were not designed as controlled ablations. In particular, the comparison between the positive replication and the follow-up near-null result involves changes in configuration beyond debater training alone, which limits how confidently we can attribute the difference to any single factor. Similarly, the Llama-based and `o4-mini` debaters were trained with different methods– DPO via self-play for Llama, and reinforcement fine-tuning through the OpenAI API for `o4-mini`. These procedures differ in their reward signals, optimization dynamics, and practical constraints (e.g., no self-play pipeline for RFT). As a result, the `o4-mini` results cannot cleanly separate the effect of using a stronger model from the effect of using a different training method.

Our approach also inherits two limitations of Arnesen et al. (2024) and some other related work: all of our judges are language models rather than humans, and our tasks instantiate the debater-judge gap mainly through information asymmetry. This made our experiments tractable and comparable to existing results, but it means that our results speak primarily to model-on-model oversight, and may be a poor proxy for settings where differences between the judge and debaters arise from misalignment rather than from capability differences alone (Bowman, 2022).

Additionally, our analysis of whether training actually improved debaters' persuasiveness is limited. Arnesen et al. (2024) computed Elo scores from full round-robin tournaments across many checkpoints and showed a monotonic positive trend between debater skill and judge accuracy. We report head-to-head win rates only for the Llama debater evaluated by GPT-4.1, and even there we did not detect a significant relationship between training duration and win probability. This makes it difficult to determine, in cases where judge accuracy did not improve, whether the debaters failed to become more persuasive or whether increased persuasiveness simply failed to help the judge.

Finally, the experiments reported here do not tell us whether unsuccessful settings reflect properties of the task, the judge, the training setup, or some interaction among these factors. Controlled experiments to tease these apart could be a promising future direction. Additionally, consultancy baselines were out of scope for this study due to resource constraints, but could have been enlightening. In particular, while Arnesen et al. (2024) showed that training did not improve judge accuracy for their strongest consultancy variant (double consultancy), it was arguably trained on an inappropriate training objective (the single-consultancy objective, which incentivizes one-sided persuasion rather than the production of arguments that are compelling when juxtaposed with opposing ones). Double consultancy, with an appropriate training objective, would be a natural baseline for comparison in future work, as would open consultancy (Roger, 2024).

## 6 Conclusion

We revisited the claim that training models to persuade judges via free-text debates can improve judge accuracy, and asked whether that effect generalizes across tasks and model classes. In a close replication of Arnesen et al. (2024), training a Llama-based debater via self-play improved GPT-4.1 judge accuracy on QuALITY. However, this effect did not replicate in a follow-up configuration. More broadly, our findings showed that this effect was not robust across all of the settings we tested. It did not extend to stronger `o4-mini` debaters trained via RL, nor to an alternative task on the Lojban language.

Our main takeaway is that while free-form debate may improve judge accuracy in some settings, it cannot be assumed to do so reliably across different tasks and training configurations. Future progress will require identifying the conditions under which debate helps judges reliably. In particular, more systematic comparisons across tasks, debaters, judges, debate protocols, and training setups will be necessary to determine whether and under what conditions debate serves as a dependable method for scalable oversight.

## Acknowledgments

Gareth Tan, Leonid Tsyplenkov, and Edy Nastase all received financial support from the Cambridge AI Safety Hub / Meridian Impact's MARS programme while completing this research. Gabriel Recchia served as a mentor with the same programme but received his financial support through a grant from Open Philanthropy (now Coefficient Giving) which also covered compute costs. Gareth Tan additionally discloses a contractual engagement with Arcadia Impact within the 36 months preceding submission. Gabriel Recchia additionally discloses contractual engagements with Anti Entropy (fiscal sponsor for grants provided through UK AISI's Challenge Fund and Alignment Fund programmes), and Hidden Variable Limited (regarding a project funded by Anthropic), within the 36 months preceding submission.

## Reproducibility Statement

We provide implementation details sufficient to reproduce the experimental setup. Sections 3 and 4 describe datasets, prompts, training procedures, and evaluation protocol. Appendix B reports the supervised fine-tuning configuration used for the Llama-based debater. Appendices C and D provide representative evaluation configurations. We identify the underlying public base models used in each experiment, describe the construction of the QuALITY and Lojban datasets, and report sample sizes for the main evaluations.

Our experiments build on the public codebase released by Arnesen et al. (2024), with task-specific modifications described in the paper. Some components of our pipeline depend on API-hosted models and fine-tuned endpoints, and the specific OpenAI models we used are likely to be deprecated at some point in the future. Nevertheless, the paper provides the information needed to reproduce the experimental design, data construction, and evaluation methodology so long as the relevant model APIs continue to be available. Code for the experiments is included in the anonymised supplementary material and can be found at https://github.com/garetht/nyu-debate-modeling.

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

# A    Lojban Task Development and Validation

Prior to deployment in the multi-round debate setting, we conducted validation experiments to ensure that the Lojban task provided appropriate difficulty and meaningful signal for AI debate evaluation. Our validation protocol consisted of two primary experimental conditions. In the *single-model evaluation* condition, a standalone language model was prompted to answer Lojban questions from the development dataset without debate interaction. In contrast, the *single-round debate baseline* was a simplified one-round debate using the prompting strategy from Arnesen et al. (2024).

These experiments were designed to establish whether the task complexity fell within an appropriate range (neither trivially simple nor prohibitively difficult) and to provide performance baselines for comparison with full multi-round debates.

## A.1    Context selection experiments

To determine how much background information to provide and how it should be selected, we evaluated multiple strategies on the 124-question development set, measuring accuracy both with and without the provided information. In this context, accuracy is defined as either producing correct answers (in single-model evaluation) or convincing judges to select correct answers (in debate settings).

### A.1.1    Performance by context strategy

Table 4: Accuracy across different context provision strategies. * indicates that this condition frequently exceeded context-window constraints.

| Configuration | Single Model | Single-Round Debate |
| --- | --- | --- |
| No background | 48% | 56% |
| Full lessons + all definitions + grammar chapters | 56% | – |
| 2 sections per word (tf-idf) + relevant definitions | 66% | 66% |
| 3 sections per word (tf-idf) + relevant definitions | – | 70%* |

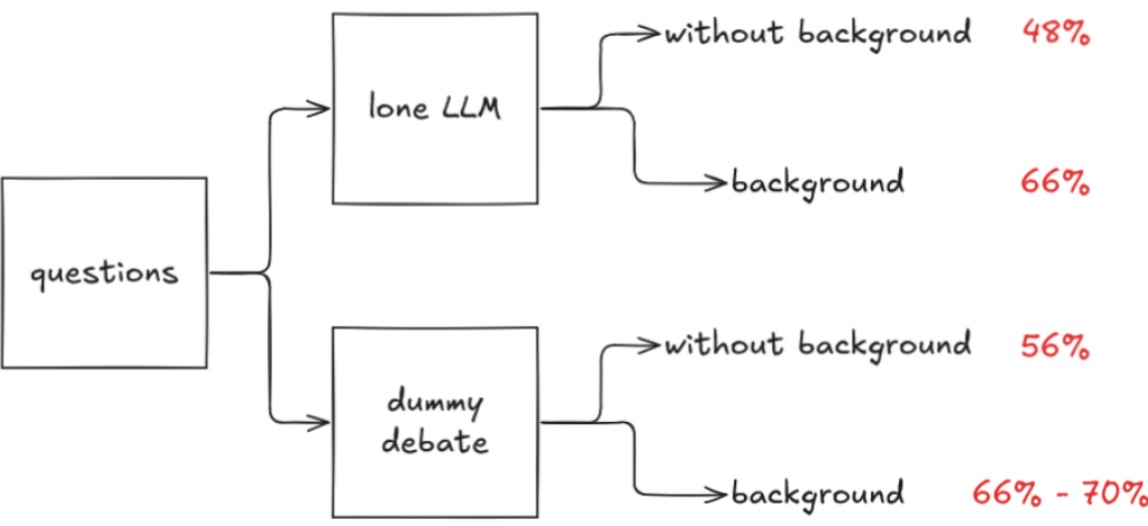

Figure 2: A visual summary of the performance achieved by different strategies on the Lojban task development set.

Retrieving two tf-idf-ranked sections per Lojban word, combined with relevant definitions, seemed to be the best strategy, as it provided an approximately 20 percentage point improvement over the no-background baseline without retrieving so many tokens as to exceed the context window.

## A.2 Implementation Details

The experimental framework builds on the codebase from Arnesen et al. (2024) with two key modifications for the Lojban task. First, background retrieval was implemented as a separate preprocessing step outside the main debate pipeline, using the tf-idf algorithm described in the main text. This avoided integration complexity and computational overhead during debate execution. Second, custom `DataLoader` and `Dataset` classes were implemented to handle the Lojban question format and associated background context.

Our validation experiments suggested that the Lojban task, with appropriately selected background context, was likely to provide a suitable testbed for AI debate evaluation, with sufficient difficulty to be informative while remaining tractable for current language models. One limitation of these analyses is that the characteristics of the questions in our development set differed substantially from those in the test set, meaning that the analyses described above may not all generalize to it. During task development, we used *gpt-4.1-nano* as a judge to assess the feasibility of the Lojban task. This model is different from the judges used in the main experiments. In these preliminary tests, *gpt-4.1-nano*, judging transcripts from trained *Llama-3-8B-Instruct-262k* debaters with access to background information, significantly outperformed the same judge evaluating transcripts from untrained debaters.

## B   SFT Training Configuration

```
Train - Llama3 - Human and GPT:
  model_name:
  /{local directory}/downloaded-models/gradientai/Llama-3-8B-Instruct-262k
  llm_type: llama3
  target: debater
  opening_speeches_only: False
  max_length: 32986
  training_hyperparameters:
    num_train_epochs: 2
    per_device_train_batch_size: 2
    gradient_accumulation_steps: 8
    optim: adamw_hf
    learning_rate: 2e-4
    max_grad_norm: 0.3
    warmup_ratio: 0.03
    lr_scheduler_type: constant
  peft_type: lora
  logging_and_saving_config:
    logging_steps: 10
  output_dir:
/{local directory}/trained_models/llama-3-mega-no-judge-speeches-31.07
  merge_output_dir:
/{local directory}/trained_models/llama-3-mega-merged-no-judge-speeches-31.07
  dataset:
    - dataset_type: quality_debates
      full_dataset_file_path:
/{local directory}/sft_data/debater/debater_combined_filled.jsonl
    - dataset_type: external_huggingface
      full_dataset_file_path: yahma/alpaca-cleaned
```

## C   Original Experiment Configuration

```
debater-sft-trained-0731-judge-trained-0731:
  batch_size: 1
  num_speeches: 2
  flip: False
  enable_self_debate: True
  speech_structure: default_debate
  alternate: False
  agents:
    debaters:
      - model_settings:
          model_type: llama3
          model_file_path: /{local directory}/trained_models/
            llama-3-mega-merged-no-judge-speeches-31.07
          alias: "dpo-rl-0731-model"
          require_quote_validation: True
          generation_params:
            temperature: 0.5
    judge:
      model_settings:
        model_type: openai
```

```
        alias: openai-judge-ft-0731
        model_file_path: ft:{anonymized-openai-judge-id}
  dataset:
    dataset_type: quality
    split_type: val

debater-round2-rl-trained-0808-judge-trained-0731:
  batch_size: 1
  num_speeches: 2
  flip: False
  enable_self_debate: True
  speech_structure: default_debate
  alternate: False
  agents:
    debaters:
      - model_settings:
          model_type: llama3
          model_file_path: /{local directory}/trained_models/
            llama-3-DPO-round-two-full-trained
          alias: "dpo-rl-round-2-model"
          require_quote_validation: True
          generation_params:
            temperature: 0.5
    judge:
      model_settings:
        model_type: openai
        alias: openai-judge-ft-0731
        model_file_path: ft:{anonymized-openai-judge-id}
  dataset:
    dataset_type: quality
    split_type: val
```

## D    Follow-up Experiment Configuration

```
eval-llama-3-262k-2025-07-31-debater_sft-only-
gpt-41-2025-07-31-judge_sft-only-quality:
  agents:
    debaters:
    - best_of_n: null
      model_settings:
        alias: llama-3-262k-2025-07-31-debater
        generation_params:
          do_sample: true
          max_new_tokens: 300
          repetition_penalty: 1.2
          temperature: 0.5
          top_p: 1.0
          use_generation_penalties: false
        is_human: false
        model_file_path: /{local directory}/trained_models/
          llama-3-mega-merged-no-judge-speeches-31.07
        model_type: llama3
        nucleus: true
        offline_file_path: null
```

```
        override_prompt: null
        peft_base_model: null
        probe_hyperparams: null
        require_quote_validation: true
        served: false
      scratchpad:
        scratchpad_public: false
        scratchpad_word_limit: null
        use_scratchpad: false
    judge:
      best_of_n: null
      model_settings:
        alias: gpt-41-2025-07-31-judge
        generation_params:
          do_sample: true
          max_new_tokens: 300
          repetition_penalty: 1.2
          temperature: 0.5
          top_p: 1.0
          use_generation_penalties: false
        is_human: false
        model_file_path: ft:{anonymized-openai-judge-id}
        model_type: openai
        nucleus: true
        offline_file_path: null
        override_prompt: null
        peft_base_model: null
        probe_hyperparams: null
        require_quote_validation: true
        served: false
      scratchpad:
        scratchpad_public: false
        scratchpad_word_limit: null
        use_scratchpad: false
  alternate: false
  annotations_classifier_file_path: null
  batch_size: 1
  convert_to_double_consultancy: false
  dataset:
    combine_train_and_val: false
    dataset_type: quality
    flip_sides: false
    full_dataset_file_path: null
    shuffle_deterministically: true
    split_type: val
    supplemental_file_paths: {}
    test_file_path: null
    train_file_path: null
    val_file_path: null
  enable_self_debate: true
  flip: false
  multi_round_branching: none
  num_speeches: 2
  previous_run: null
```

```
  prompt_config:
    default_prompt_name: Debate Prompt
    file_path: null
    hardcoded_topic_config: null
    is_memorized: false
    use_hardcoded_topics: false
  speech_structure: default_debate
  tournament:
    custom_matchups: null
    replication_file_paths: []
    tournament_type: round_robin

eval-llama-3-262k-41-sft-judge-round-2-debater_round-two-dpo-
gpt-41-2025-07-31-judge_sft-only-quality:
  agents:
    debaters:
    - best_of_n: null
      model_settings:
        alias: llama-3-262k-41-sft-judge-round-2-debater
        generation_params:
          do_sample: true
          max_new_tokens: 300
          repetition_penalty: 1.2
          temperature: 0.5
          top_p: 1.0
          use_generation_penalties: false
        is_human: false
        model_file_path: /{local directory}/trained_models/
          llama-3-DPO-round-two-full-trained
        model_type: llama3
        nucleus: true
        offline_file_path: null
        override_prompt: null
        peft_base_model: null
        probe_hyperparams: null
        require_quote_validation: true
        served: false
      scratchpad:
        scratchpad_public: false
        scratchpad_word_limit: null
        use_scratchpad: false
    judge:
      best_of_n: null
      model_settings:
        alias: gpt-41-2025-07-31-judge
        generation_params:
          do_sample: true
          max_new_tokens: 300
          repetition_penalty: 1.2
          temperature: 0.5
          top_p: 1.0
          use_generation_penalties: false
        is_human: false
        model_file_path: ft:{anonymized-openai-judge-id}
```

```
        model_type: openai
        nucleus: true
        offline_file_path: null
        override_prompt: null
        peft_base_model: null
        probe_hyperparams: null
        require_quote_validation: true
        served: false
     scratchpad:
        scratchpad_public: false
        scratchpad_word_limit: null
        use_scratchpad: false
  alternate: false
  annotations_classifier_file_path: null
  batch_size: 1
  convert_to_double_consultancy: false
  dataset:
    combine_train_and_val: false
    dataset_type: quality
    flip_sides: false
    full_dataset_file_path: null
    shuffle_deterministically: true
    split_type: val
    supplemental_file_paths: {}
    test_file_path: null
    train_file_path: null
    val_file_path: null
  enable_self_debate: true
  flip: false
  multi_round_branching: none
  num_speeches: 2
  previous_run: null
  prompt_config:
    default_prompt_name: Debate Prompt
    file_path: null
    hardcoded_topic_config: null
    is_memorized: false
    use_hardcoded_topics: false
  speech_structure: default_debate
  tournament:
    custom_matchups: null
    replication_file_paths: []
    tournament_type: round_robin
```

