# OpenReview forum: "Investigating the limits of free-form debate as a scalable oversight strategy"
_TMLR — Accepted by TMLR_

### Review · Reviewer_BUKd · 2026-05-06

**Summary Of Contributions:**

Overview: This paper re-examines the effectiveness of free-form debate for scalable oversight, with a particular focus on whether self-play debate training can consistently and generalizably improve judge accuracy. In experiments that closely replicate the original results on the QuALITY dataset, the authors find positive evidence that self-play training of Llama-based debaters can improve GPT-4.1 judge accuracy under certain conditions. However, in a near-identical follow-up setup, this effect fails to reproduce. Moreover, when extending the experiments to a stronger debater model, additional debate training actually leads to a decrease in judge accuracy. The authors further evaluate the approach on a reasoning-heavy benchmark, where no improvement from debate training is observed. Overall, the paper suggests that while free-form debate may be promising under specific conditions, its effectiveness is highly dependent on the task, model, and training setup. As such, it does not currently appear to be a robust or model-agnostic approach for scalable oversight.

Novelty / Significance:  Although the paper does not propose a new debate method itself, it has novelty in that it replicates prior key results, extends them to stronger models, and introduces a new reasoning-heavy task based on Lojban. In particular, the negative findings regarding robustness and generalization constitute a meaningful contribution to the scalable oversight literature, where positive results have been predominant and the boundary conditions of debate remain poorly understood.

Strengths
•	Rather than merely replicating prior results, the paper provides a valuable empirical replication study that additionally evaluates robustness and generalization.
•	By reporting not only positive results but also negative findings and experimental sensitivity, the work offers important insights into the reliability of debate-based scalable oversight.

Weaknesses
•	In the extension to stronger debaters, model strength and training methodology are confounded, which limits the interpretability of the observed negative results.
•	While the instability of the positive replication results is clearly reported, the lack of a systematic analysis of the factors driving this sensitivity weakens the robustness claims.

**Audience:**

Yes

**Audience Explanation:**

The paper addresses a timely and important topic in scalable oversight, empirical AI safety, and language model evaluation. Researchers interested in debate, alignment, judge reliability, and robustness of oversight methods would likely find the negative and unstable findings informative. Even if the paper does not establish definitive conclusions, its results are relevant to the broader community because they caution against treating free-form debate as a broadly reliable oversight mechanism.

**Broader Impact Concerns:**

I do not see major ethical concerns that would require rejection. However, because the paper studies scalable oversight, the broader impact discussion should explicitly address the risk of overclaiming the reliability of debate-based oversight. If debate training improves judge accuracy only in narrow or unstable settings, deploying such methods prematurely could create a false sense of safety.

**Claims And Evidence:**

No

**Claims Explanation:**

The paper provides suggestive evidence that free-form debate is sensitive to task, model, and training setup, but the evidence is not yet sufficiently convincing to support the stronger robustness-related claims. In particular, the near-identical replication failure is central to the paper, but the causes of this failure are not systematically analyzed. Similarly, the stronger-debater experiment is difficult to interpret because model strength and training procedure are confounded. The reported results are interesting and potentially important, but the current evidence does not fully disentangle the mechanisms behind the observed failures.

**Requested Changes:**

Major Comments:
•	The failure to reproduce results under an almost identical setup is not sufficiently analyzed. As shown in Section 4.1, changing only shuffle_deterministically=True leads to a performance difference (0.790 → 0.775), but the underlying factors driving this sensitivity are not systematically investigated. A deeper analysis is needed to strengthen the central replication claim.
•	In Section 4.1, increased quote usage after debate training is reported alongside improvements in judge accuracy, but it remains unclear whether quote usage is causally responsible for the improvement or merely a correlated artifact. It would be beneficial to further analyze the relationship between quote behavior and judge accuracy.
•	The transcript segmentation procedure in Section 3.1.1 generates multiple training samples that are not fully independent, potentially inflating the effective sample size and introducing correlated supervision. The implications of this data construction choice on both training and evaluation validity should be more carefully discussed.

Minor Comments:
•	Tables 1 and 2 are difficult to locate due to insufficiently clear captions. It would be better to separate key explanations into dedicated notes below the tables.
•	Sections 3.1.2 and 3.1.3 appear to overlap in content. Both discuss debate transcript formatting and preprocessing, but the distinction between them is not clearly defined.

---

> ### Author Response · Authors · 2026-05-26
>
> Thank you for the detailed critique. We agree that the original framing made the robustness claim too strong. The revised paper now states a narrower conclusion: debate training can reproduce the Arnesen et al. result in a close QuALITY setting, but the effect is fragile and should not be assumed to generalize across nearby configurations, model pairings, or tasks.
> Agreed that the near-identical replication failure needed deeper analysis. We now report that the successful 2,502-example replication was not stable across alternative 2,502-example subsets from the same larger evaluation stream. This means the instability cannot be explained solely by setting shuffle_deterministically=True; the estimated effect is sensitive to evaluation subset and ordering even when the substantive debate protocol is unchanged. We also clarify that this flag fixes the random seed used in dataset ordering, not the models, prompts, or debate format.
> We also agree that quote usage should not be interpreted causally. In the revision, quote usage is reported as an associated behavioural change after debate training, alongside the accuracy result, rather than as the mechanism responsible for the improvement. The statistical details are now moved into a compact table, and GEE is defined as generalized estimating equations.
> On transcript segmentation, we clarify in the methods that longer human transcripts were segmented into multiple shorter debates by progressively adding pairs of speeches. We agree that these examples are not fully independent; the revised limitations emphasize that more controlled experiments are needed to identify which components of the debate pipeline are responsible for observed gains.
> We also implemented the presentation fixes. The table captions are shorter, with key explanations moved into notes below the tables. The previous overlap between debate formatting and preprocessing sections has been removed by consolidating that material into a single “Debate format and preprocessing” subsection; the following subsection now concerns model selection and fine-tuning.

---

### Review · Reviewer_rDX5 · 2026-05-08

**Summary Of Contributions:**

This article replicates and expands the experiments of Arnesen et al. (2024) to (1) test the reproducibility of their findings on the "debate" method and (2) test their generalization to (a) newer and bigger (in terms of parameters) models and (b) a different benchmark. The authors replicate the original results using the original setup, but find that they do not generalize across models and benchmarks.

**Audience:**

Yes

**Audience Explanation:**

Generalization of model behavior beyond specific model selections in a project is one of the most important questions in LLM research.

**Claims And Evidence:**

Yes

**Claims Explanation:**

The paper makes three substantive claims:
1. The original experiments from Arnesen et al. (2024) successfully replicate.
2. These findings do not generalize across models.
3. These findings do not generalize across datasets.

The first claim is well supported. The replication of the original setup (as closely as possible) is thorough and well-documented, and the points of divergence are well argued. The replicated results are reasonably consistent.

The second claim is well supported. The choice of alternative models is sufficient to make such a claim, and the observed drop in accuracy both after fine-tuning and compared to a non-debate setting are clear.

The third claim is more nuanced. The observed failure to transfer the original results to the CELS-Lojban-based benchmark is technically enough to reject generalization in the strong sense. However, there are two critical points:
1. QuALITY was published before the tested models, while the Lojban dataset was released after them. This raises the concern that the former might have been present in the models training data, while the latter could not have been, which might be an explanation for the diverging results.
2. The Lojban dataset is indeed very different from the QuALITY dataset, perhaps so structurally and semantically different that any claims beyond the rejection of hard generalization are difficult to make.

On balance, the claims are sufficiently supported in aggregate.

**Requested Changes:**

Ideally, the paper would include at least one more benchmark that is closer structurally and semantically to the QuALITY dataset.

In absence of such an extension, the authors should at least add a deeper discussion of the structural and semantic differences between the datasets and their potential impact, including a discussion of whether the results could be explained by training data contamination, as QuALITY was released before the tested models while the CELS-Lojban was released after.

Further there are two claims that deserve reconsideration:
1. "[...] a weaker judge (e.g. a human [...]" in the Introduction and "[...] weaker evaluators, such as humans [...]".
Humans being weaker evaluators than LLMs, of any kind, is both a very strong claim and arguably a form of anthropomorphization that is best avoided.
2. "This creates an information asymmetry that allows the debaters to act as stand-ins for strong future models that have greater knowledge than the judges, even if the debater is not more capable than the judge in an absolute sense."
This too is a very strong claim, and raises questions such as why would you only use stronger and newer debater models and not judge models in the future.

---

> ### Author Response · Authors · 2026-05-26
>
> Thank you for the careful and constructive review. We agree that the dataset-generalization claim should be interpreted cautiously. In the revision, we have narrowed the framing: the Lojban result is presented as evidence that the QuALITY result does not automatically transfer to a structurally different task, rather than as evidence identifying the precise causal reason for that failure.
> We agree that QuALITY and CELS-Lojban differ substantially. The revised paper now makes this contrast clearer: QuALITY is a long-context reading-comprehension task where debaters can often help by selecting and quoting relevant evidence, whereas the Lojban task requires reasoning over grammatical rules and sentence-composition structure. We also note that debaters were not trained on Lojban-related data, so the Lojban results should be read as an out-of-distribution test rather than as a matched benchmark comparison.
> We also agree on the wording around “weaker” humans and future strong models. We have removed language describing humans as weaker evaluators and instead refer to judges as limited by task-relevant information, time, or domain expertise. We also revised the “stand-ins for strong future models” motivation. The QuALITY setup is now described as a controlled information-asymmetry proxy, not as a claim that debaters are globally more capable than judges or that future oversight systems should strengthen debaters without also improving judges.
> We did not add a closer third benchmark in this revision, however we agree that this would be valuable future work, especially a benchmark closer to QuALITY in structure while still testing reasoning beyond evidence selection.

---

### Review · Reviewer_S7Xn · 2026-05-14

**Summary Of Contributions:**

The authors are studying the effect of debate, which in this context is a method for producing training data for a judge model.  Two strong models with access to source texts argue opposing sides of a question, and through their reasoning and quotation of the original text, they provide training data for a relatively weaker judge, even in the case where the original text is withheld from the judge.

The authors were particularly interested in seeing if results from a previous work by Arnesen et al could be replicated under similar conditions but with stronger models for all roles involved, and where the questions under debate are more complicated: the models are asked to reason about the syntactic validity of claims in a constructed language (Lojban), requiring employing definitions and arguments rather than finding and extracting spans of definitive text.

While they were able to replicate the original finding, they could not do so with regularity, finding that multiple runs of same conditions as Arnesen et al lead to ver different outcomes, only occasionally producing the reported judge accuracy.  Overall, I read this as a negative result on the question of whether debate is a scalable and robust method for improving judge accuracy.

**Audience:**

Yes

**Audience Explanation:**

Despite the esoteric use of a constructed language, I think some TMLR readers would be interested to learn that debate is limited as a means of judge improvement.

**Broader Impact Concerns:**

None.

**Claims And Evidence:**

No

**Claims Explanation:**

To the extent that they were able, the results suggest that the original experiments with Llama family of models improved judge accuracy in debates (cf. sections 3.1.1 - 3.1.6).

### The biggest issue: experimental designs are full of confounds

In section 3.1.6, the authors describe a two-round DPO regime for improving the judge.  But they don't specify if Arnesen et al used two rounds of DPO, or if they only used one; could they be more precise about that here?

In 3.2.1, the authors describe a very different training construction to the original setting for QuALITY.  This sounds like it departs from the scenario previously about context size.  Do the authors think it might contribute to the failure to reproduce the Arnesen results?  Another difference comes up in 3.2.2 with the scalar reward design.  They don't mention if they tried to validate whether the probability was calibrated or not, or to try and characterize how noisy the reward signal was.  This contributes to suspicion that these differences confound the ability to test if debate can extend to stronger models.

This same issue comes up again in 4.1, where the authors mention casually that they are using head-to-head win probability, whereas Arnesen used average tournament Elo.  These are very different measures, and it seems impossible to tell if their study is under-powered due to too few debates per checkpoint, or if training for a longer duration really has an effect on debater performance.

With so many departures from the original experiments, it is difficult to conclude that the negative result is truly sound or if it is due to one of the many differences introduced.

**Requested Changes:**

In section 3.1.6, the authors describe a two-round DPO regime for improving the judge.  But they don't specify if Arnesen et al used two rounds of DPO, or if they only used one; could they be more precise about that here?

In 3.2.1, the authors describe a very different training construction to the original setting for QuALITY.  This sounds like it departs from the scenario previously about context size.  Do the authors think it might contribute to the failure to reproduce the Arnesen results?  Another difference comes up in 3.2.2 with the scalar reward design.  They don't mention if they tried to validate whether the probability was calibrated or not, or to try and characterize how noisy the reward signal was.  This contributes to suspicion that these differences confound the ability to test if debate can extend to stronger models.

This same issue comes up again in 4.1, where the authors mention casually that they are using head-to-head win probability, whereas Arnesen used average tournament Elo.  These are very different measures, and it seems impossible to tell if their study is under-powered due to too few debates per checkpoint, or if training for a longer duration really has an effect on debater performance.

In 4.1, the authors discuss fitting a population-averaged GEE, but to not define this intialism, I think they should.  In the same paragraph, they describe in words the outputs in a set of sentences, which I think would be more effectively communicated by a figure or table
```
smaller but still significant effect (β = 0.184, SE = 0.061, z = 3.03, p = .002; OR = 1.20, 95% CI [1.07,
1.35]). We also found that this training increased the use of quoted evidence. For the gpt-4.1-based judge,
using OLS with cluster-robust standard errors by item, the average number of valid quotes per debate rose
from 11.75 before training to 18.69 after training (SE of difference= 0.185, t = 37.56, p < .001; 95% CI
for difference [6.58, 7.31]). Invalid quotes increased from 0.44 to 0.74 (SE= 0.047, t= 6.53, p<.001; 95%
CI [0.21, 0.40]), and total quotes rose from 12.18 to 19.43 (SE= 0.196, t = 37.08, p < .001; 95% CI [6.87,
7.63]). This experiment utilized 2,502 samples and the configuration reported in Appendix C.
```

The following paragraph begins with a rather ambiguous sentence:
> However, this result did not replicate in a subsequent run with a configuration that differed only in that all values were explicitly specified to the values that the original configuration defaulted to, with the exception of shuffle_deterministically, which was changed to True.

Do the authors mean that they only changed `shuffle_deterministically` to True (whatever this means), and the rest of the configuration was unchanged?

Overall, I am worried that this inability to extend to a ‘stronger’ model is because the experimental conditions cannot be sufficiently controlled by fine-tuning an open AI model via API calls.

Could the team not have tried to fine-tune a more performant (but similarly sized) open weight model, say from the Qwen family?  The motivation for trying debate on a stronger model makes sense, but the motivation to jump to a completely different experimental setup, and to relinquish SFT control to an API is not.

---

> ### Author Response · Authors · 2026-05-26
>
> Thank you for the careful review. We agree that several parts of the paper needed to distinguish more clearly between a close replication and broader generalization tests.
> We have clarified the DPO setup in Section 3.1.5. DPO was applied to the debater, not to the judge; the judge supplied the preference signal but its parameters were not updated during DPO training. We also now state explicitly that Arnesen et al. used two DPO iterations, and that our close replication followed the same two-round structure.
> We agree that the training regime in 3.2.1 differs and that this makes it difficult to diagnose whether this or debater strength is the greater contributer to the failure to reproduce the results. However, given that we observed reproduction failures for even much more similar configurations, these results support our conclusion that the free-form debate should not be assumed to reliably improve judge accuracy across different tasks and training configurations. We used head-to-head win probability rather than average tournament Elo due to compute constraints; it did suggest the debaters in the main replication improved with training (Sec 4.1), despite the limitations discussed in 5.1.
> We also agree that the o4-mini experiment is not a clean causal ablation of model strength alone. The revised limitations make this explicit: Llama debaters were trained with DPO via self-play, while o4-mini was trained through the OpenAI RFT API, so reward signals, optimization dynamics, and practical constraints differ. We therefore frame the o4-mini result as evidence that the debate-training effect did not survive this practically relevant stronger-debater setup, not as proof that debater strength itself caused the failure.
> Finally, we revised Section 4.1 to define GEE, move the dense statistical results into a table, and clarify the shuffle_deterministically issue. The follow-up run fixed dataset ordering and made inherited defaults explicit; it did not intentionally change the debate protocol, models, or prompts.

---

### Decision · Action_Editor_hKFb · 2026-06-13

**Recommendation:** Accept as is

**Audience:**

Yes

**Audience Explanation:**

A multi-agent debate is a technique where multiple AI agents are assigned different perspectives, hypotheses, or solution strategies and then critique each other's reasoning before producing a final answer. Debate is an emerging framework and will be interested by TMLR's audience.

**Claims And Evidence:**

Yes

**Claims Explanation:**

This paper revisits the effectiveness of debate as a scalable oversight mechanism for AI systems. The authors successfully replicated a previously reported positive result on the QuALITY benchmark but found that the effect is fragile, does not consistently reproduce under near-identical conditions, and fails to generalize to stronger models or a reasoning-intensive Lojban benchmark.

The paper does not show that debate is ineffective, but it does provide evidence that the benefits of free-form debate are fragile and may not generalize reliably across tasks, models, and training settings. This is a valuable insight and is weakly supported with some experimental evidence. The claims of the paper have been appropriately narrowed following the reviewers' feedback.